# Chronic Obstructive Pulmonary Disease Is Associated with Worse Oncologic Outcomes in Early-Stage Resected Pancreatic and Periampullary Cancers

**DOI:** 10.3390/biomedicines11061684

**Published:** 2023-06-10

**Authors:** Rachel Huang, Emma Hammelef, Matthew Sabitsky, Carolyn Ream, Saed Khalilieh, Nitzan Zohar, Harish Lavu, Wilbur B. Bowne, Charles J. Yeo, Avinoam Nevler

**Affiliations:** 1Sidney Kimmel Medical College, Thomas Jefferson University, Philadelphia, PA 19107, USA; rachel.huang@jefferson.edu (R.H.); emma.hammelef@jefferson.edu (E.H.);; 2Jefferson Pancreas, Biliary and Related Cancer Center, Department of Surgery, Thomas Jefferson University, Philadelphia, PA 19107, USA

**Keywords:** PDAC, pancreatic ductal adenocarcinoma, chronic obstructive pulmonary disease, COPD, oncologic outcome, prognostic markers, tumor microenvironment

## Abstract

Pancreatic ductal adenocarcinoma (PDAC) is the 3rd leading cause of cancer mortality in the United States. Hypoxic and hypercapnic tumor microenvironments have been suggested to promote tumor aggressiveness. The objective of this study was to evaluate the association between chronic obstructive pulmonary disease (COPD) and oncologic survival outcomes in patients with early-stage PDAC and periampullary cancers. In this case-control study, patients who underwent a pancreaticoduodenectomy during 2014–2021 were assessed. Demographic, perioperative, histologic, and oncologic data were collected. A total of 503 PDAC and periampullary adenocarcinoma patients were identified, 257 males and 246 females, with a mean age of 68.1 (±9.8) years and a mean pre-operative BMI of 26.6 (±4.7) kg/m^2^. Fifty-two percent of patients (N = 262) reported a history of smoking. A total of 42 patients (8.3%) had COPD. The average resected tumor size was 2.9 ± 1.4 cm and 65% of the specimens (N = 329) were positive for lymph-node involvement. Kaplan–Meier analysis showed that COPD was associated with worse overall and disease-specific survival (*p* < 0.05). Cox regression analysis showed COPD to be an independent prognostic factor (HR = 1.5, 95% CI 1.0–2.3, *p* = 0.039) along with margin status, lymphovascular invasion, and perineural invasion (*p* < 0.05 each). A 1:3 nearest neighbor propensity score matching was also employed and revealed COPD to be an independent risk factor for overall and disease-specific survival (OR 1.8 and OR 1.6, respectively; *p* < 0.05 each). These findings may support the rationale posed by in vitro laboratory studies, suggesting an important impact of hypoxic and hypercapnic tumor respiratory microenvironments in promoting therapy resistance in cancer.

## 1. Introduction

Pancreatic ductal adenocarcinoma (PDAC) is a highly aggressive neoplasm and is the most prevalent type of pancreatic malignancy, accounting for more than 90% of pancreatic cancers [1]. PDAC is associated with a five-year overall survival rate of only 12% and high recurrence rates, leading to it being the third leading cause of cancer death in the United States [2].

In recent years, extensive research has focused on understanding the impact of the pancreatic tumor microenvironment on cell proliferation, metabolic reprogramming, chemotherapy resistance and immune evasion [3,4,5]. In particular, the cancer’s respiratory microenvironment has been shown to play a crucial role in these processes though hypoxia-related signaling and other pathways. The hypoxia signaling pathway is now a novel target of cancer research, being assessed in multiple clinical trials, and with Belzutifan, a HIF2α inhibitor being recently FDA-approved for VHL-driven cancers [6]. Another often-overlooked aspect of the respiratory microenvironment of many cancers is the concomitant elevation of carbon dioxide levels (i.e., hypercapnia). Hypercapnia has been shown to promote cancer aggressiveness and chemoresistance in lung cancer, colon cancer and pancreatic cancer [7,8,9,10,11]. We have also recently shown that gene expression signatures extracted from pancreatic cancer cells exposed to hypercapnia can be applied in patient tissue specimens and used as a strong prognostic marker for disease-specific survival [7]. This, coupled with previous reports of hypercapnia-related resistance to platinum chemotherapy [8], highlights the importance of the respiratory microenvironment in cancer advancement and treatment response.

Chronic obstructive pulmonary disease (COPD), the fourth leading cause of death in the United States, is characterized by physiologic changes that promote a hypercapnic and hypoxic tissue environment. Interestingly, large studies have shown that COPD is associated with higher rates of lung, pancreas, colon, liver, breast, prostate and esophageal cancers [12,13]. Specifically, as it relates to pancreatic cancer, Kornum et al. [12] and Chiang et al. [14] have reported increased rates of pancreatic cancer in individuals with COPD. A possible indication of the mechanism may be gleaned from the recent study of Ream et al. [15] that suggested COPD as being associated with pancreatic intraductal papillary mucinous neoplasms in males.

The precise pathophysiological mechanisms linking COPD and pancreatic adenocarcinoma remain unclear. It has been shown that the chronic inflammatory nature of COPD is linked to the presence of inflammatory mediators that contribute to the development of lung cancer [16]. The rise in oxidative stress observed in COPD has also been proposed to exacerbate COPD disease progression and increase susceptibility to exacerbations as well as other comorbidities [17]. It is possible that the COPD microenvironment—characterized by hypercapnia, hypoxia, the abundance of inflammatory mediators and oxidative stress [17]—fosters a pro-inflammatory environment that promotes the development and progression of PDAC; especially, since chronic and recurrent pancreatitis has been identified as a risk factor for PDAC. It may also be possible that COPD and PDAC share pathophysiological mechanisms simply because they share common risk factors such as smoking, age, and environmental exposures. It becomes important to understand the association between COPD and PDAC because it can help, clinically, with risk assessment for individuals with COPD and help tailor preventive treatment as well as help guide treatment decisions and patient management.

Previously, we have also shown COPD and other obstructive respiratory conditions to be important prognostic factors for survival in early-stage, resectable pancreatic cancer [8]. These observations were noted in a cohort of non-active smokers that underwent surgical resection between 2004 and 2014 [8]. Since then, there has remained a question about the generalizability and the relevance of those findings in the era of multi-agent drug combination therapies.

In light of the evolving data regarding the importance of the respiratory microenvironment in tumor progression, our study aimed to assess the role of COPD as a negative prognostic factor in early stage PDAC in a contemporary, larger and robust patient population, as investigating the association between COPD and PDAC serves not only to enhance our understanding of these diseases but also has important implications in assessing patient risk and in managing targeted therapy.

## 2. Materials and Methods

### 2.1. Data Collection

This was a retrospective observational study with a case-control design. The dataset was based on a prospectively maintained single-institutional database of patients who underwent a curative-intent pancreaticoduodenectomy for pancreatic cancer at the Jefferson Pancreas, Biliary, and Related Cancer Center between 2014 and 2021. Patient records were reviewed through the electronic medical system and demographic, perioperative, pathohistological and oncologic outcome data were collected. Charts were reviewed for completeness of data, and for evidence of chronic respiratory disorders, including COPD, asthma and obstructive sleep apnea (OSA). Both COPD and asthma were further stratified into steroid-independent or steroid-dependent subgroups.

### 2.2. Data Collection and Cohort Selection

Data were reviewed from a prospectively maintained IRB-approved pancreatic surgery database of patients who underwent pancreaticoduodenectomy at the Jefferson Pancreas, Biliary and Related Cancer Center between 2014 and 2021. Data were collected on patients’ smoking history, past medical history, pathology and clinical outcomes, among other information. Electronic medical records were reviewed to determine smoking history and identify patients who had a documented diagnosis of COPD or OSA. Included cases were composed of patients older than 18 years old with a histological diagnosis of PDAC or other periampullary cancer (ampullary, bile duct and duodenal cancers). Patients that were found to have metastatic cancer during surgery were excluded from the analyses. Non-smokers, current smokers and past smokers were all included. Overall, a total of 863 patient records were reviewed and a total of 503 early-stage, resectable pancreatic cancer patients were identified and included in the final study cohort.

### 2.3. Statistical Analysis

Student’ *t*-test and Mann–Whitney test were used to compare continuous variables for parametric and non-parametric variables, respectively. The χ^2^ test was used to compare categorical variables. Survival was assessed with Kaplan–Meier and Cox regression analyses. The Cox regression model was used to assess the impact of significant factors on the overall and disease-specific survival. The model was further optimized by sequential exclusion of statistically insignificant factors (*p* ≥ 0.2) until achievement of a final optimal model fit (*p* ≤ 0.05). Propensity score matching was performed with a 1:3 no-replacement nearest-neighbor matching for age, cancer type, lymphovascular invasion, perineural invasion, resection margin status, lymph node metastasis and preoperative neoadjuvant treatment. *p* values ≤ 0.05 were considered statistically significant. Statistical data were analyzed with IBM SPSS (Version 28.0.1.0). The propensity score matching was performed in the R language and environment for statistical computing, version 4.2.0 (22 April 2022 ucrt), using the “survival”, “MatchIt” and “survminer” R packages.

### 2.4. Ethical Approval

This study was performed at Thomas Jefferson University Hospital (TJUH) and approved by the local institutional review board. The data supporting the findings of this study are available from the corresponding author upon request.

## 3. Results

### 3.1. Patient Cohort Characteristics

The initial cohort included 863 patients who underwent a pancreaticoduodenectomy procedure at Thomas Jefferson University Hospital. Of that population, 503 patients with PDAC and periampullary adenocarcinoma were identified and included in our final cohort (Table 1). Our study cohort consisted of 257 males and 246 females, with a median age of 68 (29–95) years and a median pre-operative BMI of 25.8 kg/m^2^ (17.5–49 kg/m^2^). There were 262 patients (52%) with a history of smoking and of these, 215 (42.7%) reported quitting prior to surgery. In our cohort, 109 individuals had a chronic respiratory disorder—42 patients (8.3%) were diagnosed with COPD, 41 (8.2%) with asthma, 48 (9.5%) with OSA and 19 patients (3.8%) were diagnosed with multiple respiratory disorders. Most of the patients in our cohort were diagnosed with pancreatic ductal adenocarcinoma with the rest being diagnosed with periampullary cancer (85% versus 15%). The average resected tumor size was 2.9 ± 1.4 cm, and lymph node metastasis was found in 329 (65.4%) patients. Overall survival data were available for all the patients. Recurrence data were based on electronic medical records availability.

### 3.2. COPD Status Does Not Correlate with TNM Staging or Histopathological Risk Factors

Overall, the COPD and non-COPD groups were similar in age, sex, racial distribution and BMI. Not surprisingly, the COPD groups had a considerable association with a current or previous history of smoking (*p* < 0.01). TNM staging, lymphovascular invasion and perineural invasion were not associated with COPD status. Though not significant, the non-COPD group had higher rates of neoadjuvant treatment (22.1% vs. 9.5%, *p* = NS). In the PDAC tumor subgroup, this was significant (25.3% vs. 10.5%, *p* = 0.042). A deeper inspection of the COPD cohort, with stratification based on therapy (chronic steroid user vs. non steroid users), found that COPD patients who were chronic steroid users (inhaled or systemic) had significantly smaller tumors (2.1 ± 0.8 vs. 3.2 ± 1.4 cm, *p* = 0.04), as seen in Appendix A. Though not statistically significant, COPD patients who were chronic steroid users were also less likely to be active smokers, and presented with less lymph node metastasis, lower rates of lymphovascular and perineural invasion and lower rates of surgical tumor positivity.

### 3.3. OSA Status Does Not Correlate with TNM Staging

Obstructive sleep apnea was more prevalent in male patients (OR 2.5, 95% CI 1.3–4.83, *p* = 0.006) and correlated with higher pre-operative BMI (AUC 0.665, 95% CI 0.578–0.751, *p* < 0.001). Overall, the OSA and non-OSA groups were similar in age, and racial distribution. OSA did not correlate with TNM staging, surgical margin status, or perineural invasion. Though only trending toward significance, OSA was associated with higher rates of lymphovascular invasion (OR 1.8, 95% CI 1.0–3.5, *p* = 0.065). No differences were observed in neoadjuvant and adjuvant therapy rates between patients in the OSA group and the non-OSA group. Of note, pre-operative morbid obesity (BMI > 35 kg/m^2^) also had a statistical trend toward higher lymphovascular invasion (OR 2.6, 95% CI 1.0–6.7, *p* = 0.056).

### 3.4. Survival Analysis Reveals COPD to Be a Negative Risk Factor for Overall Survival

The overall survival of our cohort was 29.6 ± 1.8 months and the median disease-free survival (DFS) was 18.9 ± 1.4 months. For PDAC patients, the overall survival was 26.5 ± 1.8 months and the DFS was 17.4 ± 0.9 months. COPD was associated with a significant decrease in median overall survival (22.4 ± 5.4 vs. 30.5 ± 1.8 months, *p* = 0.024) in PDAC and periampullary cancers, as shown in Figure 1. Kaplan–Meier analysis also showed tumor margin positivity (18.9 ± 3.1 vs. 31.8 ± 2.0 months, *p* < 0.001), the presence of perineural invasion (26.17 ± 2.0 vs. 87.9 ± 23.8 months, *p* < 0.001), BMI < 20 kg/m^2^ (19.5 ± 7.6 vs. 30.4 ± 1.9 months, *p* < 0.05) and lymphovascular invasion (25.1 ± 2.2 vs. 39.8 ± 6.1 months, *p* < 0.001) to be significant risk factors for worse overall survival. In a PDAC-specific analysis, COPD was again shown to be significantly associated with poor survival (22.4 ±5.7 vs. 27.93 ± 1.9 months, *p* = 0.026) as shown in Figure 2. Surprisingly, smoking history was not found to be significantly associated with median survival (26.4 ± 2.7 vs. 31.1 ± 2.3, *p* = NS). Disease-specific survival (DSS) showed a COPD to trend toward poorer survival for the entire cohort (PDAC and periampullary cancer) with a median survival of 32.2 ± 6.1 months vs. 39.8 ± 4.3 months (*p* = 0.065). Specifically, in the PDAC group, COPD was found to be more significantly associated with survival with a median DSS of 24.2 ± 4.9 months vs. 33.7 ± 3.1 months, as seen in Figure 2.

Analysis of disease-free survival did not show COPD as a significant factor (11.7 (9.4–14.1] months vs. 12.7 (11.5–14.8) months, *p* = NS).

In a multivariate Cox regression: tumor margin positivity, the presence of perineural invasion (PNI), lymphovascular invasion (LVI) and a history of COPD were all found to be significant and independent risk factors for poor overall survival (*p* < 0.001 for the model) as shown in Figure 3. In a PDAC-specific analysis of overall survival, tumor margin positivity, LVI, PNI and adjuvant chemotherapy were all found to be independent risk factors for worse survival, with COPD showing similar effects, though with borderline statistical significance (Appendix A).

In a Cox regression analysis of DSS, tumor margin positivity, LVI, PNI and adjuvant chemotherapy were all found to be independent risk factors for worse DSS in PDAC, as shown in Appendix A. COPD was found to be similarly impactful as in overall survival, though it did not reach statistical significance (HR 1.45, *p* = 0.115). Cox regression of DSS in periampullary cancers was unable to identify any significant risk factors.

After 1:3 nearest neighbor propensity score matching for age, cancer type (PDAC vs. periampullary cancer), lymphovascular invasion, perineural invasion, tumor resection margin status, lymph node metastasis and preoperative neoadjuvant treatment, two cohort matches were created (COPD = 35 patients and Control = 105 patients). Regression analyses were performed for both overall survival and disease-free survival (See Table 2). COPD status was found to be independently associated with overall survival (HR 1.8, 95% CI 1.1–2.8, *p* = 0.012) and DSS (HR 1.6, 95% CI 1.0–2.5, *p* = 0.045).

### 3.5. COPD Treatment-Dependent Subpopulations Correlate with Survival

Kaplan–Meier analyses revealed that COPD patients not chronically treated with steroids have worse overall survival compared to non-COPD patients (16.4 [5.0–27.8] months vs. 30.5 [27.0–34.0], *p* = 0.035). Similarly, these patients also had worse disease-specific survival (24.9 [12.3–37.5] months vs. 50.3 [40.2–60.3], *p* = 0.005). Kaplan–Meier survival analysis of COPD patients who received chronic treatment with steroids failed to show any significant differences in overall or disease-specific survival when compared to non-COPD controls. Cox regression analysis of overall survival and disease-specific survival was performed including the COPD treatment-dependent subpopulations (Table 3). The Cox regression found the treatment type of COPD to be a borderline independent risk factor for disease-specific survival but not for overall survival. As expected, steroid-dependent COPD patients correlated with worse hazard ratios compared with non-steroid dependent COPD patients.

### 3.6. Survival Analysis Reveals OSA to Be a Possible Risk Factor for Overall Survival

Overall, median survival was not different between the OSA and non-OSA groups in the study cohort (29.9 vs. 29.4 months, respectively. *p* = NS). For non-PDAC periampullary tumors, OSA patients had a trend towards worse overall survival (39.3 ± 14.8 vs. 66.8 ± 5.3 months, respectively. *p* = 0.08). Surprisingly, this finding was not observed in PDAC patients (40.2 ± 6.9 vs. 38.9 ± 2.0, respectively. *p* = NS). In a multivariate Cox regression, BMI, tumor size, lymph node involvement and perineural invasion were not found to be contributing to the regression model. The final optimal model (*p* = 0.0003) included only tumor margin positivity (HR 20.4, 95% CI 2.3–183.6, *p* = 0.007) and OSA status (HR 2.6, 95% CI 0.9–7.7, *p* = 0.078). Assessment of disease-specific survival did not reveal OSA to be associated survival (*p* = NS). OSA status was not found to be a significant independent prognostic factor for survival in PDAC (*p* = NS). Similarly, in a 1:3 propensity score matched analysis (OSA = 35 patients vs. Control = 105 patients), OSA was not found to correlate with overall or disease-specific survival.

## 4. Discussion

Pancreatic cancer is an aggressive disease with a poor prognosis, thus necessitating continued research interest. Currently, it is estimated that 62,000 individuals in the United States will be newly diagnosed with pancreatic cancer annually [18]. As one of the most prevalent and deadly cancers, pancreatic cancer has an overall five-year survival of only 12% and is estimated to be the second most common cause of cancer deaths by 2030 [2,18,19]. As a result, efforts have been made to identify factors that aid in attenuating the drastic consequences of pancreatic cancer.

Current identified prognostic factors in pancreatic cancer survival include tumor-staging characteristics such as size [20] and resectability [21], patient physiologic factors such as frailty and/or sarcopenia [22,23], and molecular markers including KRAS, BRCA, TP53, ADAM9 [9] and MSI [24] status. Extensive research has gone into investigating the tumor microenvironment as a driver of tumor aggression. More than a century after Paget’s “seed and soil” hypothesis, we know now that the ‘soil’ is a composed of a complex interaction between basement proteins, stromal cells, immune cells with their secreted factors, involved microbiome and the resulting metabolome of that diseased tissue. As the metastatic tumor selects and shapes its surrounding environment—thereby, optimizing conditions for growth and survival—it is essential to study the different aspects of the microenvironment as a key to developing and selecting effective strategies [25,26,27].

The respiratory tumor microenvironment is an essential component of the tumor biology. Hypoxia and activation of the hypoxia-inducible-factor 1A (HIF1-A) pathway is a major effector in tumorogenesis, tumor therapy resistance, epithelial to mesenchymal transition, invasion and metastasis [28,29]. Furthermore, hypoxic tissue signatures have been shown to be closely associated with poor survival outcome in patients with pancreatic cancer [30]. Similarly, hypercapnia has also been associated with tumor aggressiveness, therapy resistance [8,9,10] and immunosuppression [31,32]. It is therefore possible to theorize that diseases of the respiratory system that present with systemic hypoxia and/or hypercapnia may confer additional risk through their effects on the tissue microenvironment.

COPD is a prevalent disease and a leading cause of disability and death in the United States, affecting over 15 million Americans today [33]. It is the fourth leading cause of death in the United States and the third worldwide. In the United States, tobacco smoking is the leading cause of non-inherited COPD. However, it is also important to note that one in four individuals diagnosed with COPD will have never smoked, suggesting that this disease can also be caused by environmental toxins, air pollutants [34,35] or respiratory infections [36] coupled with some genetic predisposition in certain individuals (e.g., alpha 1-antitrypsin deficiency). It is primarily composed of two respiratory pathologies: chronic, recurrent bronchitis, and emphysema. Kornum et al. have found a strong association between individuals with COPD and increased rates of pancreatic cancer [12], and Ream et al. [15] have recently reported an increased incidence of IPMN in patients with COPD. These support the potential role of COPD as a causative factor in pancreatic cancer. Unsurprisingly, COPD shares many risk factors with lung, oropharyngeal, esophageal, pancreatic, colon and gastric cancer development [12,13,14,37,38]—with smoking being the predominant factor, followed closely by air pollution [39]. In a study by Shah et al., COPD was found to be significantly associated with decreased survival in non-small cell lung cancer (NSCLC). This effect was noted to be more pronounced in early-stage NSCLC (stages I–II) [40]. Similarly, Chen et al. showed that patients with colon cancer who also had an acute exacerbation of COPD prior to cancer treatment experienced a worse prognosis—even more so if their exacerbation resulted in hospitalization [38]. This broad effect of COPD on survival, in the context of multiple cancers, is potentially suggestive of a systemic effect of COPD rather than loco-regional interaction.

Our study has shown that COPD is associated with a worse overall and disease-specific survival in patients with early-stage, resectable PDAC, thereby validating a finding that was previously shown by Nevler et al., in a separate cohort comprised of patients who were treated between 2002 and 2014 (14.9 ± 2.3 vs. 19.8 ± 1.1 months, *p* < 0.05) [8]. Furthermore, these new results were observed within a carefully matched group of patients, ensuring comparable tumor staging and invasive histological markers across the cohort. We also attempted to assess the effect of COPD severity on cancer survival. Although Global Initiative for Chronic Obstructive Lung Disease (GOLD) [41] classification was not recorded for our patients, medication data were available. We therefore subdivided the COPD patients group based on chronic steroid use, using this as a possible surrogate marker for moderate–severe disease. Surprisingly, though not statistically significantly, COPD patients receiving steroid therapy (inhaled and/or oral corticosteroids) tended to present at earlier cancer stages and with smaller tumors (*p* = 0.04). This corresponded with better overall survival and disease-specific survival. However, when accounting for disease stage and pathological risk factors such as staging, the presence of lymphovascular invasion or surgical margins containing tumor cells, the survival of these patients tended to be worse than the other COPD patients (though not in a significant amount).

This contemporary analysis validates our previous observations regarding obstructive respiratory diseases as prognostic risk factors in early-stage, resectable PDAC. Importantly, this new report which looks at a cohort of patients from the last decade succeeds in accounting for implementation of multi-agent treatment protocols, thereby improving the generalizability of previous observations. More specifically, while the previous study assessed patients who were operated on between 2002 and 2014 and primarily treated with single-agent adjuvant therapy [8], the patient cohort in this study includes mostly patients treated primarily with multi-agent protocols. Additionally, the cohort inclusion criteria differed in that our current 2014–2021 cohort also included individuals who reported smoking at the time of surgery, whereas in Nevler et al., the 2002–2014 cohort only comprised non-smokers and former smokers. Within our current cohort of patients with COPD, our results showed that there were no significant differences in survival when comparing patients with and without a history of smoking and when comparing current smokers (at the time of surgery) to patients who quit prior to surgery—a finding that was also found by Wang et al. in lung cancer [11].

We believe that several interconnected mechanisms are involved in COPD’s deleterious effects in pancreatic cancer survival. First, COPD is a classic example of a hypercapnic and hypoxic respiratory pathology. These conditions, as mentioned, result in a clear aggressive cancer phenotype. Hypoxia as a stressor has the capability to activate hypoxia-inducible factors (HIFs), triggering signaling pathways that promote cellular proliferation, stimulating angiogenesis and inducing metabolic reprogramming [28,29,42,43]. Furthermore, the presence of hypercapnic and hypoxic environments, as is seen in COPD, hinders the expression of NF-κB, thereby disrupting the function of the innate immune system [31,32,44,45,46], which may potentially enable PDAC cells to evade the immune system. Escribese et al. found that inducing hypoxic environments, results in pronounced predominance of the M2 phenotype of tumor-associated macrophages (TAMS) [45]. These M2 TAMS contribute to tissue remodeling/repair and angiogenesis [46], thereby promoting tumor proliferation. Furthermore, lymphocytes, specifically cytotoxic T-cell (CD8+), were significantly downregulated in individuals with COPD, even when controlling for smoking status [47]. These processes collectively aim to optimize the cellular microenvironment to support robust proliferation.

We have also attempted to study the effect of obstructive sleep apnea in our cohort. OSA is a disease characterized mostly by nocturnal, recurrent, transient hypoxia, and there are multiple studies suggesting OSA as a risk factor for carcinogenesis, cancer progression and increased cancer mortality [48,49,50,51,52,53]. Prior studies by Brenner et al. [52] and Palamaner et al. [53] found an increase in cancer incidence in patients with OSA. Similarly, Ream et al. found OSA to be associated with IPMN, a common precursor to PDACs if untreated, in females [15]. Our results, however, only suggested a borderline association with overall survival (HR 2.6, *p* = 0.078) and were unable to show any significant impact on disease-specific survival. Similarly, our propensity score matched analysis was unable to find significant associations of OSA with survival. These results coincide with the 2016 findings of Dal Molin et al. [51] who analyzed the prognostic impact of OSA in a cohort of PDAC patients from Johns Hopkins Hospital between 2003 and 2014. Therefore, while we find the results regarding COPD quite compelling, we are cautious about the impact of OSA in our specific patient cohort.

While our study did not focus directly on the association between respiratory diseases and their effects on the tumor microenvironment, we can theorize that a possible crosstalk between the dysregulated respiratory physiology and the pancreatic cancer microenvironment is likely. While further studies are required to explore this direction—if true—the clinical implications are quite notable. Should resolution of hypoxia or hypercapnia be considered as clinical pulmonary goals of care in patients receiving chemotherapy, and should systemic and/or tissue hypoxia/hypercapnia be measured to inform on treatment success? Would aggressive respiratory therapy synergize with adjuvant chemotherapy in these setting?

We hope that as the research progresses in this field, especially with HIF pathway inhibitors already being assessed in clinical trials, these questions will soon be answered.

## 5. Limitations

Our study had several limitations. This was a retrospective study, and as such, is exposed to selection and misclassification biases. As this was based off a prospectively maintained, cross-validated database, we believe that we have been able to substantially mitigate some of those biases. Secondly, the different groups also may vary in presentation. Although, this is partially addressed by the propensity score matching, it is still a potential weakness of the study. It is also possible that the underlying respiratory diseases somehow affected or limited the future treatments, thereby affecting survival. While this is not a limitation of a weakness per se, but rather an alternative proposed mechanism, it is important to note that in our patient group, COPD did not pose a major limitation to adjuvant chemotherapy as the multi-drug protocols are rarely associated with pulmonary toxicity. The cohort only included cases of PDAC and periampullary cancers in patients who underwent resections of the pancreatic head; therefore, it may be limited in its generalizability to cases involving advanced-stage PDAC and in resectable cancers of the pancreatic body and tail. Second, while our study only included a single institution, it is now the second largest study to detect these findings. Third, given the retrospective nature of our study, some data may be missing due to incomplete charts or patients being lost to follow-up. Fourth, our data capture relied upon patient or referring physician reporting of smoking history or respiratory disease diagnosis in the electronic medical record.

## 6. Conclusions

Smoking and chronic obstructive pulmonary disease are associated with an increased risk of pancreatic cancer. Our study validates COPD as an important prognostic factor for worse overall survival in patients with early-stage pancreatic cancer. We therefore believe that COPD status should be considered in post-resection treatment and decision-making. Attempts to prevent acute exacerbations of COPD—such as implementing COPD maintenance therapy, encouraging smoking cessation and recommending annual flu vaccinations—need to be considered as oncologic adjuncts in PDAC therapy. We are also optimistic that future studies of the hypercapnic/hypoxic environment may identify novel strategies (chemotherapeutics, targeted radiation, other) that may lead to improved patient outcomes.

## Figures and Tables

**Figure 1 biomedicines-11-01684-f001:**
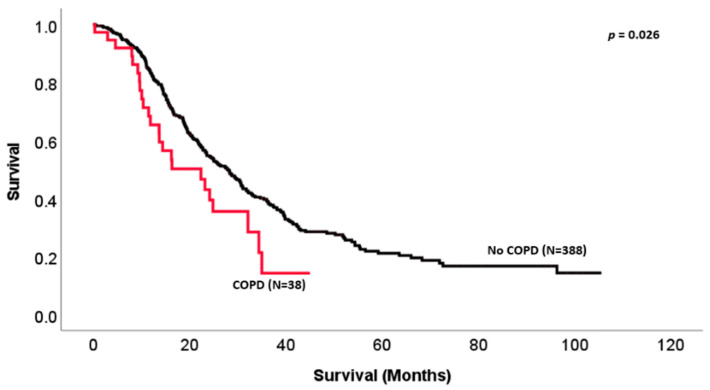
Kaplan–Meier overall survival curve of early-stage PDAC patients alone who underwent pancreaticoduodenectomy comparing overall survival in patients with chronic obstructive pulmonary disease (COPD) to patients with no medical history of COPD. (Median survival 22.4 ± 5.7 vs. 27.9 ± 1.9 months, *p* = 0.026.)

**Figure 2 biomedicines-11-01684-f002:**
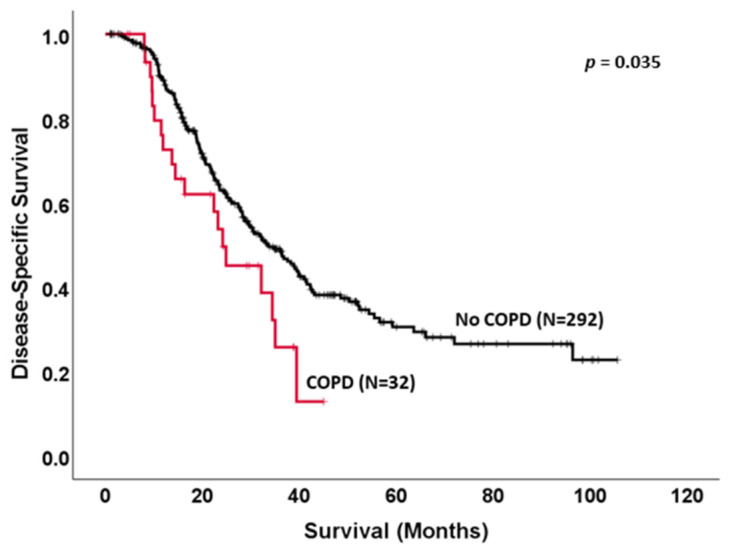
Kaplan–Meier survival curve of early-stage PDAC patients alone who underwent pancreaticoduodenectomy comparing disease-specific survival (DSS) 0 in patients with chronic obstructive pulmonary disease (COPD) to patients with no medical history of COPD (24.2 ± 4.9 vs. 33.7 ± 3.1 months, *p* = 0.035).

**Figure 3 biomedicines-11-01684-f003:**
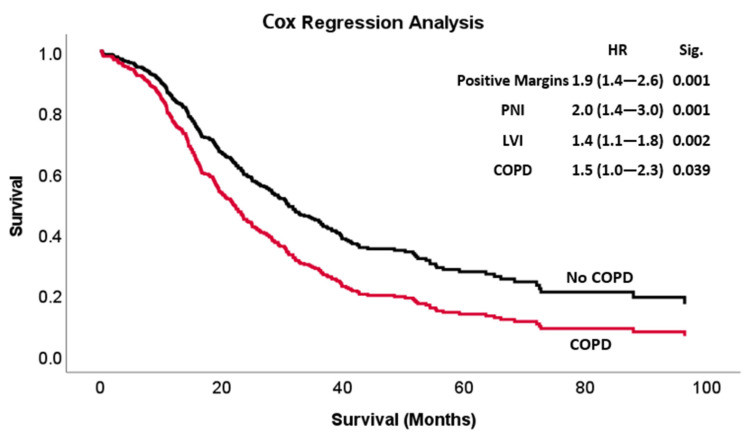
Multivariable Cox regression curve (*p* < 0.001) of early-stage PDAC and periampullary patients who underwent pancreaticoduodenectomy, comparing overall survival in patients with COPD and patients with no medical history of COPD. COPD—chronic obstructive pulmonary disease, PNI—perineural invasion, LVI—lymphovascular invasion.

**Table 1 biomedicines-11-01684-t001:** Tumor and patient characteristics (n = 503).

Characteristic	Overall (n = 503)	No COPD (n = 461)	COPD (n = 42)	*p*-Value
Age, y, median (IQR)	68 (29–95)	68 (29–95)	69 (53–86)	NS
Sex, n (%)				NS
Male	257 (51%)	236 (51%)	21 (50%)	
Female	246 (49%)	225 (49%)	21 (50%)	
BMI (range), kg/m^2^	25.8 (17.5–49)	25.8 (17.5–49)	25.5 (19.1–36.6)	NS
Race, n (%)				NS
White	418 (83.1%)	382 (82.9%)	36 (85.7%)	
Black	47 (9.3%)	42 (9.1%)	5 (11.9%)	
Asian	15 (2.9%)	14 (3.0%)	1 (2.4%)	
Other	16 (3.1%)	16 (3.5%)	0 (0%)	
Unknown	7 (1.4%)	7 (1.5%)	0 (0%)	
Smoking History, n (%)				
Nonsmoker	241 (47.9%)	235 (51%)	6 (14.3%)	**0.001**
Past Smoker	215 (42.7%)	190 (41.2%)	25 (59.5%)	**0.022**
Current Smoker	47 (9.3%)	36 (7.8%)	11 (26.2%)	**0.001**
Neoadjuvant Chemotherapy	106 (21%)	102 (22.1%)	4 (9.5%)	NS
Tumor type, n (%)				
PDAC	426 (84.6%)	388 (84.2%)	38 (90.5%)	NS
Periampullary	77 (15.3%)	73 (15.8%)	4 (9.5%)	NS
Tumor Size, mean (cm)	2.9 ± 1.4	2.9 ± 1.4	2.9 ± 1.4	NS
Lymph Node Involvement, n (%)	329 (65.4%)	300 (65.1%)	29 (69.0%)	NS
Positive Margins, n (%)	59 (11.7%)	53 (11.5%)	6 (14.3%)	NS
LVI, n (%)	266 (52.9%)	244 (52.9%)	22 (52.4%)	NS
PNI, n (%)	408 (81.1%)	370 (80.3%)	38 (90.5%)	NS
Tumor Staging, n (%)				
T				NS
T1	62 (12.3%)	58 (12.6%)	4 (9.5%)	
T2	152 (30.2%)	137 (29.7%)	15 (35.7%)	
T3	272 (54.1%)	249 (54.0%)	23 (54.8%)	
T4	13 (2.6%)	13 (2.9%)	0 (0%)	
TX	4 (0.8%)	4 (0.9%)	0 (0%)	
N				NS
N0	170 (33.8%)	158 (34.3%)	13 (31.0%)	
N1	268 (53.3%)	247 (52.6%)	21 (50.0%)	
N2	63 (12.5%)	55 (11.9%)	8 (19.0%)	

COPD—chronic obstructive pulmonary disease, IQR—interquartile range, PDAC—pancreatic ductal adenocarcinoma, LVI—lymphovascular invasion, PVI—perineural invasion, NS—nonsignificant. **Bolded** values indicate statistical significance.

**Table 2 biomedicines-11-01684-t002:** Cox regression analysis of 1:3 propensity score matched cohort (Total = 140 patients: COPD = 35 patients and Control = 105 patients), assessing prognostic factors for overall survival and disease specific survival. Regression model significance *p* < 0.001. COPD—chronic obstructive pulmonary disease; HR—hazard ratio.

	Factor	HR	*p*-Value
**Overall survival**	Lymphovascular invasion	1.1 (1.01–1.18)	0.024
	Lymph node metastasis	1.6 (0.96–2.81)	0.070
	Tumor margin positivity	2.5 (1.47–4.19)	<0.001
	COPD	1.8 (1.13–2.79)	0.012
**Disease-specific survival**	Lymphovascular invasion	1.1 (1.01–1.18)	0.028
	Lymph node metastasis	1.7 (0.96–3.01)	0.067
	Tumor margin positivity	2.5 (1.42–4.54)	<0.001
	COPD	1.6 (1.01–2.54)	0.045

**Table 3 biomedicines-11-01684-t003:** Cox regression analyses assessing prognostic factors for overall survival and disease specific survival including COPD and the corresponding COPD treatment-type. Regression model significance *p* < 0.001. COPD—chronic obstructive pulmonary disease; HR—hazard ratio. * compared to non-COPD patients.

	Factor	HR	*p*-Value
**Overall survival**	Lymphovascular invasion	1.4 (1.06–1.81)	0.016
	Perineural invasion	2.2 (1.45–3.41)	<0.001
	Tumor margin positivity	1.7 (1.24–2.41)	0.001
	Neoadjuvant treatment	1.38 (1.01–1.88)	0.046
	Tumor size (cm)	1.10 (0.99–1.20)	0.068
	COPD *		0.328
	Non-steroid dependent	1.3 (0.82–2.11)	0.264
	Steroid Dependent	1.5 (0.67–3.48)	0.308
**Disease-specific** **survival**	Lymphovascular invasion	1.4 (1.04–1.98)	0.029
	Perineural invasion	3.1 (1.78–5.50)	<0.001
	Tumor margin positivity	1.6 (1.08–2.46)	0.019
	Neoadjuvant treatment	1.92 (1.37–2.70)	<0.001
	COPD *		0.058
	Non-steroid dependent	1.7 (0.97–2.94)	0.066
	Steroid Dependent	2.1 (0.84–5.13)	0.110

## Data Availability

The data presented in this study are available upon request from the corresponding author. The study data are not publicly available because they are part of a large prospectively maintained database containing genetic data.

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
