# Peer review of "Chronic Obstructive Pulmonary Disease Is Associated with Worse Oncologic Outcomes in Early-Stage Resected Pancreatic and Periampullary Cancers"

_biomedicines, 2023, doi:10.3390/biomedicines11061684_

Round 1

Reviewer 1 Report

1. Nothing is said about the severity of COPD, there is no gradation on the GOLD scale. COPD was not in the acute phase? take the drugs took place?

2. There is a typo in figure 1: should the red curve be labeled COPD?

3. Patients with COPD are all current or past smokers, was this factor also taken into account in the multifactorial analysis? COPD can be caused by smoking.

4. Nothing is said about treatment in the COPD/no COPD groups. Maybe patients with COPD have treatment restrictions and therefore lower survival rates?

5. Why did the authors choose to have pancreatic cancer, for example, in lung cancer, the prevalence of COPD is much higher?

Reviewer 2 Report

The article entitled “Chronic Obstructive Pulmonary Disease is Associated with Worse Oncologic Outcomes in Early-Stage Resected Pancreatic 3 and Periampullary Cancers” is well-written and, from my point of view, woul be of interest for the readers of Biomedicines. In spite of this and before its publication I would recommend authors to perform the following changes:

In the introduction the layout of the manuscript must be described.

Also, in the introduction the aims of the manuscript should be clearly stated.

Line 62: it is said ‘This was a retrospective case-controlled’ is it correct? For me it would be better to say case-control

In 2.2. Data Collection, a more in-depth description of where data come from would be convenient.

Table 1 and 2: I recommend substituting Sig and Significance by p-value. The same can be applied to Table 3.

English is good enough for a scientific journal.

Reviewer 3 Report

The paper is well written. The methods and results are presented in depth. Some changes need to be made to the other parts before possible publication.

ABSTRACT enter the paper's objectives.

INTRODUCTION The introduction provides a clear overview of pancreatic ductal adenocarcinoma (PDAC) as a highly aggressive neoplasm and its impact on patient survival. However, it would be beneficial to further elaborate on the relationship between chronic obstructive pulmonary disease (COPD) and PDAC. Specifically, the authors should consider providing a more detailed explanation of the potential pathogenetic mechanisms linking COPD to PDAC and discussing the clinical implications of this association. Furthermore, in the introduction, it would be helpful to provide a brief summary of the existing literature on the role of the tumor microenvironment, particularly the respiratory microenvironment, in PDAC. This could include discussing the significance of hypoxia-related signaling pathways and the emerging research on the impact of hypercapnia on cancer aggressiveness and chemotherapy resistance. Considering the relevance of COPD as a risk factor for various cancers, including PDAC, it would be valuable to mention any previous studies that have investigated the association between COPD and other cancer types. This would enhance the context and highlight the significance of studying COPD in relation to PDAC. Lastly, the authors can clearly state the objective of their study, which is to validate and reevaluate the prognostic role of COPD in early-stage PDAC using a larger and more contemporary patient population.

 DISCUSSION While the existing discussion touches upon the link between BPCO and cancer, it would be valuable to provide a more detailed explanation of the potential mechanisms driving this association. By elucidating the underlying pathogenic pathways and molecular interactions involved, the authors can better establish the biological rationale and strengthen the link between BPCO and the development of this particular cancer. Furthermore, to enhance the clinical relevance of the study, the authors should provide insights into the practical implications of the BPCO-cancer association. Discussing how this relationship may impact disease management, diagnosis, treatment strategies, or patient outcomes will provide valuable information for clinicians and researchers. Overall, addressing these points will significantly improve the manuscript by providing a more comprehensive understanding of the BPCO-cancer association and its clinical implications

Good

Round 2

Reviewer 1 Report

I have no more comments on the article. I believe that in its present form the manuscript can be recommended for publication.

Author Response

Thanks.

Reviewer 3 Report

The authors seem to have responded to my requests. Unfortunately, the changed parts are not highlighted in the text and therefore it is impossible to evaluate the differences with the previous version.

Author Response

Thank you for your comments, we have uploaded the manuscript with traces of revisions, please download the attachment to view.

Round 3

Reviewer 3 Report

The authors arranged the paper according to my requests